# Cellulose-Based Nanofibers in Wound Dressing

**DOI:** 10.3390/biomimetics10060344

**Published:** 2025-05-23

**Authors:** Abdul Razak Masoud, Zeinab Jabbari Velisdeh, Mohammad Jabed Perves Bappy, Gaurav Pandey, Elham Saberian, David K. Mills

**Affiliations:** 1Molecular Science and Nanotechnology, Louisiana Tech University, Ruston, LA 71272, USA; 2Micro and Nanoscale Systems Engineering, Louisiana Tech University, Ruston, LA 71272, USA; jabedperves@gmail.com; 3School of Biological Sciences, Louisiana Tech University, Ruston, LA 71272, USA; gauravpandey226005@gmail.com; 4Klinika and Akadémia Košice Bacikova, Pavol Jozef Šafárik University, 04001 Košice, Slovakia

**Keywords:** cellulose, wound, nanofiber, drug delivery, tissue engineering

## Abstract

Wound dressings have a significant role in managing trauma-related injuries, chronic lacerations, as well as post-operative complications, by preventing infections and promoting tissue regeneration. Conventional methods using sutures and gauze often pose constraints in healing effectiveness and cost. Emerging materials, particularly cellulose-based nanofibers, offer a favorable choice due to their biodegradability, biocompatibility, and structural similarity to the extracellular matrix. Cellulose, being an abundant, naturally available biopolymer, forms the basis for modern materials for wound dressing. It is a very resourceful material due to its capability to be processed into films, fibers, and membranes with tailored properties. Surface modification of cellulose membranes with nanoparticles or bioactive compounds assists in enhancing the antimicrobial properties and supports sustained drug release, essential in chronic wound infections. Electrospinning and other modern fabrication techniques allow for controlling the fiber morphology and drug-delivery characteristics. This review highlights the properties, fabrication techniques, surface functionalization, and biomedical applications of cellulose-based materials in wound care. With increasing demand for effective and cost-effective wound treatments, cellulose nanofibers stand out as a sustainable, multifunctional platform for cutting-edge wound dressings, offering improved healing, reduced scarring, and potential for amalgamation with several drug delivery and tissue engineering approaches.

## 1. Introduction

Wound dressings are topical materials applied to the surface of wounds to enhance the healing process by reducing healing times and preventing complications through bacterial infections [1]. The earliest forms of wound treatment were described five millennia ago, with improvements on ancient treatment techniques passed on from generation to generation [2]. The introduction of antibiotics and the concept that moist wound environments facilitate faster healing have revolutionized wound-healing research. Before the use of antibiotics, honey was widely used to treat wounds [3]. Other antique wound dressings included mixtures of animal fat, clay, plants, and herbs. The choice of plant wound dressing was driven mainly by the appearance of the leaves rather than scientific proof of their wound-healing properties. The use of cotton, wool, and gauze on the surface of wounds was also reported, with the selection of these items likely occurring through trial and error. Others, such as urine, dung, or blood, probably had ritualistic significance for their use in wound treatment [4], although a present study was carried out in rats to compare the wound-healing properties of urine, povidone-iodine solution, and antiseptic agents. The results showed that topical or oral administration of urine led to increased wound-healing activity. This research was based on the fact that there is evidence of the traditional use of urine in India for the treatment of wounds [5]. There are also reports of the use of honey in wound dressings being revived because of clinical evidence of its wound-healing properties [6]. Others, such as the use of sterile maggots and negative pressure for the prevention of bacterial growth and good sorption of wound exudates, have been reported [7].

Wound dressings have seen a major transition from the past. In the 19th century, researchers realized that wound areas needed to be sterilized before suturing to prevent sepsis. Hand washing before caring gradually became a common practice. These changes brought about significant decreases in the scale of wound infections [7]. Over the years, dehydration, increased inflammation, and the formation of granular tissue have been identified as a few of many factors that directly affect wound healing [8]. To address these factors, current advancements in material and tissue engineering require that wound dressings be biodegradable, sterile, and pyrogen-free, prevent infections, enable gas exchange, ensure optimal wound-surface temperature, and have a good sorption capacity concerning wound exudates. Improving the characteristics of wound dressings will cut costs, reduce pain, and prevent the formation of scars [9].

Polymers are the most suitable materials that could potentially be engineered to possess most of the properties of an ideal wound dressing. The smartness of a polymer is defined as its ability to respond to biochemical and biophysical stimuli such as pH, temperature, enzymes, magnetic force, and light [10,11,12,13,14].

## 2. Chemical Structure of Cellulose

Cellulose and its derivatives are one of the most common natural organic materials. Cellulose, the most abundant biopolymer in nature, is an integral part of the plant cell wall, algae, fungi, lichens, and some microorganisms. Structure-wise, cellulose is a linear polymer chain consisting of glucose monomers with a flat ribbon-like conformation, joined by β-1,4 linkages [15].

The formation of covalent bonds between the C-4 OH group of one glucose unit and the C-1 carbon atom of another glucose unit provides cellulose with such unique properties as biodegradability, hydrophilicity, chirality, and the ability to be chemically modified. This natural polymer typically has four main polymorphs: cellulose I, II, III, and IV. Of these, cellulose I is native and a natural polymorph existing in two forms: Iα and Iβ. Upon heat application, cellulose Iα is irreversibly converted to cellulose Iβ [16]. Cellulose II is obtained from native cellulose by recrystallization. It can be formed in two ways: (a) regeneration, which is the process of dissolving cellulose I in a solvent, followed by re-precipitation; (b) mercerization—the process of swelling native fibers in concentrated sodium hydroxide. Cellulose III is obtained from cellulose I or II by treatment with liquid ammonia or certain amines, thus resulting in Cellulose III1 and Cellulose III2, respectively. Cellulose IV1 and cellulose IV2 can be obtained from the corresponding forms of cellulose III1 and III2 by heating in glycerin to a temperature of 260–280 °C [17].

## 3. Preparation of Cellulose

The presence of hydrogen bonds in cellulose makes it practically insoluble. One way of dissolving cellulose polymer is by using non-reactive solvents so that dissolution occurs through intermolecular interactions. These solvents could be aqueous or non-aqueous. Aqueous non-reactive solvents for cellulose are solutions of inorganic salts or complex compounds, which are widely used for cellulose regeneration. The most common solvents of this group are Schweitzer’s reagents, specifically cuprammonium hydroxide (Cuam—[Cu(NH_4_)](OH)_2_) and cupriethylenediamine hydroxide (Cuen—[Cu(NH_2_CH_2_CH_2_NH_2_)_2_]OH_2_). Another aqueous cellulose solvent is an aqueous solution of Ni (tren) (OH)_2_ [tren = tris (2-aminoethyl) amine], in which the mechanism of cellulose dissolution is considered as a homogeneous esterification reaction with a resultant formation of carboxymethyl cellulose [18,19,20]. An overview of carboxymethyl cellulose is illustrated in Figure 1.

For non-aqueous solvents, special attention has been drawn to melted hydrates of inorganic salts. The most effective of this series is LiClO_4_.3H_2_O, which allows for the obtainment of a clear cellulose solution within minutes. Other non-aqueous non-reactive cellulose solvents include oxides of tertiary amines, DMSO-containing solvents, liquid ammonia [20], and sodium bases [21] or ammonium [22]. These two groups of solvents demonstrate the polysaccharide’s physical dissolution without a chemical reaction, i.e., without deriving hydroxyl groups [23].

Another method of dissolving cellulose polymer is by using reactive solvents or hydrolytically unstable, organic soluble cellulose intermediates. In this method, dissolution occurs in combination with the formation of esters and acetate derivatives. Reactive solvents are characterized by the ability of the derivative formed to further break down into cellulose through medium change (for example, changing from a non-aqueous solution to an aqueous solution) or by changing the pH of the medium [24]. Some shortcomings of this method include the toxicity of the solvents, the occurrence of side reactions during dissolution, and the formation of indefinite structures, thus making results not reproducible. Common reactive solvents used for dissolving cellulose include water, aqueous NaOH solution, ethanol, ether, acetone, and ethyl acetate [21,24].

## 4. Formation of Films and Fibers from Cellulose

Cellulose, an abundant and renewable biopolymer, presents versatile opportunities for creating films and fibers with diverse applications across industries. Techniques have been developed to leverage the unique properties of cellulose, each offering distinct advantages and applications [25,26,27]. These techniques include electrospinning, casting, solution blowing, phase separation, 3D printing, and wet spinning.

Electrospinning: Electrospinning involves applying a high voltage to a polymer solution or melt, resulting in the formation of ultrafine fibers through the electrostatic repulsion of charged droplets. This method enables precise control over fiber morphology, including diameter and porosity, making it suitable for applications such as tissue engineering scaffolds, filtration membranes, and drug-delivery systems. However, solvent toxicity and limited scalability hinder its widespread adoption for large-scale production [28,29].Casting: Casting, also known as solution casting, is a widely used technique for producing cellulose films by casting solutions onto substrates and allowing them to dry. This method offers simplicity, scalability, and control over film thickness and properties. Nevertheless, challenges related to solvent recycling, film uniformity, and limited morphological control persist, influencing its applicability in various fields such as packaging, biomedical implants, and barrier coatings [30,31].Solution Blowing: Solution blowing entails extruding a cellulose solution through a spinneret while simultaneously subjecting it to a high-velocity gas stream, leading to fiber formation. This technique offers high production rates and control over fiber morphology, making it suitable for applications requiring tailored properties, such as filtration membranes and tissue engineering scaffolds [32,33].Phase Separation: Phase separation techniques use controlled phase separation of cellulose solutions to induce film or fiber formation. By manipulating parameters such as solvent composition and temperature, precise control over film morphology and porosity can be achieved. This method finds applications in membrane filtration, tissue engineering, and controlled release systems [34].3D Printing: 3D printing involves layer-by-layer deposition of cellulose-based inks to create complex three-dimensional structures [35]. This additive manufacturing technique offers customizable designs and precise control over architecture and composition, making it ideal for fabricating custom-designed scaffolds for tissue engineering and regenerative medicine applications [36].Wet Spinning: Wet spinning utilizes coagulation baths to solidify cellulose solutions or dispersions extruded through spinnerets, forming fiber. This technique offers scalability and control over fiber properties such as diameter, mechanical strength, and surface chemistry, making it suitable for applications in textiles, composites, and biomedical devices [2].

Determining the most suitable technique for forming cellulose-based films and fibers for wound dressing applications hinges on specific requirements. Among these methods, electrospinning stands out as highly efficient thanks to its capability to produce ultrafine fibers with exceptional surface area and porosity, which is ideal for tissue engineering applications. Despite challenges such as solvent toxicity, ongoing research endeavors aim to refine parameters and expand their utility. Moreover, cellulose’s innate biocompatibility renders it an excellent choice for wound dressing materials. Leveraging techniques like electrospinning and casting, cellulose-based films and fibers can be tailored to expedite wound-healing processes [37,38,39]. For instance, electrospun cellulose nanofibers emulate the extracellular matrix, fostering tissue regeneration, while casting techniques establish an optimal moist environment crucial for angiogenesis [40].

To dissolve cellulose and its derivatives, a mixture of copper and concentrated ammonia (Schweitzer reagent) is commonly used, which assists in forming copper-ammonia cellulose fibers. These fibers are mostly used in the textile industry and are rarely used for technical purposes due to their inadequate strength. Additionally, copper-ammonia fibers are more expensive than viscose fibers, thus there has been a rapid decline in their production. Viscose fiber and cellophane are obtained after cellulose undergoes multiple processing stages: conversion to basic cellulose using NaOH, compression and crushing leading to depolymerization, filtration, acidification, and fiber formation, rapid coagulation in sulfuric acid, cross-linking, washing, and bleaching (Figure 2) [26]. The processes of obtaining viscose fibers and cellophane are identical except for the higher alpha-cellulose content when fabricating cellophane compared to viscose fiber.

In conclusion, cellulose-based materials present innovative solutions for advancing wound care and enhancing patient outcomes [41,42]. Figure 3 describes the process flow of cellulose-based film and fiber-formation techniques, highlighting key methods such as electrospinning, casting, solution blowing, phase separation, 3D printing, and wet spinning, leading to applications in biomedical, textiles, packaging, and composites.

## 5. Surface Functionalization of Cellulose Membranes

Surface functionalization is a pivotal strategy for optimizing cellulose membranes in wound dressing applications, enabling precise tailoring of their properties through strategic adjustments to surface chemistry and structure. This process involves introducing bioactive agents, polymers, or nanoparticles onto the membrane surface using both covalent and non-covalent bonding strategies [41]. Techniques such as chemical modification, biomolecule immobilization, surface, plasma treatment, or nanoparticle functionalization are employed to customize cellulose membranes, enhancing their biocompatibility, antimicrobial activity, and overall efficacy in wound healing [42,43,44,45]. Among the various techniques available, chemical modification and biomolecule immobilization have emerged as prominent strategies owing to their efficiency and versatility [42,43].

Chemical Modification

Chemical modification techniques involve the covalent attachment of functional groups onto cellulose membrane surfaces. Methods such as esterification, etherification, oxidation, or polymer chain grafting enable precise control over surface properties. This allows for introducing specific functional groups, such as carboxyl, hydroxyl, or amino groups, facilitating tailored interactions with cells, proteins, and other biomolecules [46]. Chemical modification is a well-established and efficient method, offering fine-tuning capabilities crucial for wound-healing applications [46].

II.Biomolecule Immobilization

Biomolecule immobilization entails attaching bioactive molecules directly onto cellulose membrane surfaces. This technique enables the incorporation of growth factors, antimicrobial agents, or signaling molecules, thereby enhancing membrane functionalities. Whether through physical adsorption, covalent bonding, or affinity-based interactions, biomolecule immobilization offers versatility in promoting cell adhesion, proliferation, and antimicrobial activity—essential for wound-healing processes [45,46,47]. Studies have demonstrated the promising efficacy of this approach, making it a valuable strategy in surface functionalization for wound dressing materials.

These techniques, whether employed alone or in combination, showcase significant potential in improving the therapeutic efficacy of cellulose membranes for wound-healing applications. Continued advancements in surface functionalization technologies hold promise for further optimizing cellulose-based wound dressings, offering innovative solutions to enhance patient care and outcomes [48,49].

## 6. Cellulose Membranes in Wound Healing

A popular design technique for extending the life of wound dressings and lowering the risk of chronic wound infection is the prolonged release of antimicrobial medicines. For the treatment of skin ulcers and burn wounds, cellulose-based membranes have recently emerged as the go-to alternative for wound dressings [50]. Cellulose membranes, including interconnected micro- and nano-sized pore patterns were reported to have an enhanced ability to absorb water, resulting in a moist microenvironment that promotes wound healing [51]. Since cellulose-based nanofibers have a high surface area, are biocompatible, and have superior mechanical qualities, they have become a viable material for use in wound dressing applications. Cellulose membranes’ surface functionalization can aid in enhancing certain characteristics, including mechanical strength and water repellency [52].

A novel method for cellulose membrane surface functionalization is presented, utilizing heterogeneous “click” grafting of zwitterionic sulfobetaine. With the use of this technique, zwitterionic functional groups could be added to the membrane surface, giving it superior biocompatibility and antifouling qualities. The potential application of such functionalized membranes in wound dressings to inhibit bacterial colonization and encourage wound healing exists [30]. Furthermore, the diverse strategy for surface functionalization of nanofibers, including membranes made of cellulose, for enhanced biomedical applications is emphasized. It is possible to tailor particular capabilities like antibacterial activity, controlled drug release, and improved cell adhesion for better wound-healing results by adding bioactive chemicals, nanoparticles, or conducting polymers onto the membrane surface [31].

In addition, the importance of modifying the surface of bacterial cellulose for biomedical applications has always been emphasized. Through techniques such as chemical grafting, plasma treatment, or biomolecule immobilization, the biocompatibility and bioactivity of cellulose membranes can be enhanced, making them suitable for wound dressing materials with enhanced healing properties [32].

Surface functionalization of cellulose membranes offers a flexible method for creating sophisticated wound dressings with customized qualities to satisfy the unique demands of wound care. Researchers are making significant progress in the field of biomaterials for wound-healing applications by utilizing cutting-edge surface modification techniques. These techniques hold promise for enhancing patient outcomes and healthcare practices.

Various cellulose derivatives, such as methylcellulose (MC), hydroxypropyl methylcellulose (HPMC), hydroxyethyl cellulose (HEC), hydroxypropyl cellulose (HPC), sodium carboxymethylcellulose (NaCMC), and ethyl cellulose (EC), have been explored for their role in wound dressing formulations. Their interaction with different skin layers is depicted in Figure 4. These derivatives contribute to moisture regulation, antimicrobial effects, and controlled drug release, making them valuable in wound-healing applications [50,51].

## 7. Cellulose Derivatives

In the field of membrane science and technology, the creation of innovative polymeric composites based on cellulose derivatives is an exciting and difficult study area [33]. Cellulose and its derivatives, as biocompatible polymers, have garnered significant interest for their potential applications in the biomedical field because of their appropriate mechanical and physical characteristics. Utilizing the hierarchical structure of cellulose, high specific strength, flexibility, and usefulness are naturally produced [34,35]. It also has a low density, is affordable, and decomposes naturally [36]. The porosity and interconnectivity that are desirable for biomedical applications can be tuned with cellulosic materials [2].

One of the most extensively studied membrane technology derivatives is cellulose acetate (CA). The effect of adding cellulose nanofibers to cellulose acetate membranes is consistently emphasized. The addition of nanofibers has demonstrated the ability to improve membrane characteristics like permeability and mechanical strength, thereby broadening the range of applications for CA membranes in filtration and separation processes [37]. To increase the value or adaptability of cellulose, many cellulose derivatives have been produced and applied in the biomedical industries by chemical treatment or functionalization. The functionalization pattern along the polymer chain affects the characteristics of cellulose derivatives in addition to the kind and level of substitution [38].

Furthermore, the use of cellulose derivatives in wound dressing applications has shown promise. Because of its special qualities, including its high capacity for retaining water and biocompatibility, bacterial cellulose is a good choice for encouraging tissue regeneration and wound healing [39]. Cellulose derivatives have also been extensively investigated for the development of advanced wound dressings. To meet the many needs of wound care, these dressings provide a variety of functions such as moisture regulation, antibacterial activity, and increased healing characteristics [40].

In conclusion, membranes based on cellulose derivatives provide a flexible foundation for a range of filtration and separation uses. Researchers are still working to build functional membranes with improved performance, sustainability, and durability through careful selection of cellulose derivatives and method optimization. These initiatives have a lot of potential to solve problems in environmental cleanup, healthcare, and water purification.

## 8. Nanoparticle Additions for Added Functionalities

Recent times have seen the advent of drug-resistant microorganisms having noteworthy antibiotic resistance. In order to combat such advanced microbes, scientists have started using nanoparticles in wound-healing materials, especially cellulose-based membranes. The major reason for this addition is to impart antimicrobial functionalities to the cellulose membranes. The nanoparticles additionally also possess the ability to deliver different antibiotic drugs necessary during the wound-healing process. Over the years, different types of metal, metal oxide, and polymeric nanoparticles with low in vivo toxicity have been added to the cellulose-based wound-healing materials [41]. Even if there is some cytotoxicity of any nanoparticle towards eukaryotic cells, it can be reduced by preparing their micro and nano-formulations using natural polymers such as cellulose [42].

In the case of metal nanoparticles, silver nanoparticles (AgNPs) have remained the most favored choice of researchers to be incorporated in the cellulose-based wound-healing materials, mainly because of their excellent antibacterial properties. The mechanism behind the antibacterial nature of AgNPs is through the destruction of bacteria’s cell walls and cell membranes, DNA damage, damage to the electron transport chain, inhibition of cytoplasmic and membrane proteins, as well as inactivation of bacterial enzymes and their metabolism [43]. Wu et al. incorporated the antibacterial activity in the bacterial cellulose (BC) nanofibers using self-assembled AgNPs. The performance of the composite formed (AgNP-BC) was evaluated against *Staphylococcus aureus*, and it was observed that a significant inhibition zone (3.46 mm diameter) was present in the culture plate containing the AgNP-impregnated BC nanofibers. The wound-healing effect was checked upon a rat model, and it was found that the composite helped in decreasing the inflammation along with promoting the scald wound healing [44]. The same research group also studied the antibacterial ability of AgNP-BC against *Escherichia coli* and *Pseudomonas aeruginosa* and observed a 90% reduction in the growth of these bacteria in the presence of the composite [45].

In another study by Pal et al., BC was modified with green-synthesized AgNPs using photochemical deposition for wound-healing purposes. The antibacterial activity of the developed composite was studied using the disk diffusion method, and it was observed that the composite had significant antibacterial activity against Gram-negative bacteria [46]. Singla et al. impregnated AgNPs into cellulose nanocrystals extracted from bamboo leaves to develop nanocomposite dressings in the form of films and ointments for accelerating the wound-healing process. It was observed that wounds treated with these dressings had better antibacterial activity, along with early vasculogenesis, lesser inflammation, and increased collagen content at the wounded site in mice [47]. Apart from AgNPs, copper and gold nanoparticles (CuNPs and AuNPs, respectively) have also been used in wound dressings to incorporate antibacterial properties into the wound-healing material. A group of researchers incorporated CuNPs into BC and evaluated the activity of thus formed composites against *Staphylococcus aureus, Bacillus subtilis, Escherichia coli, Candida albicans,* and *Pseudomonas aeruginosa*. The composite showed excellent antibacterial activity, and the mechanism behind this was the formation of Cu-peptide complex, which caused cell death due to reactive oxygen species (ROS) formation [48]. Another study proved the long-term antibacterial activity (up to 90 days) of CuNP-embedded BC against Gram-positive and Gram-negative bacteria, which could be an excellent property for a wound dressing material [49]. BC has also been modified with AuNPs for developing composites to treat wounds infected with Gram-negative bacteria. It was also observed that the developed composite had better efficacy compared to the antibiotics (sulfamethoxazole and cefazolin) against Gram-negative bacteria [50].

Metal oxide nanoparticles have also been explored by researchers as antibacterial agents in wound dressings. Hamedi and Shojaosadati used zinc oxide nanoparticles (ZnO NPs) as an antibacterial agent in the nanocomposite prepared using BC and schizophyllan [51]. It was observed that the presence of these nanoparticles reduced the concentration of *Escherichia coli* and *Staphylococcus aureus* by 90% and 70%, respectively. The mechanism behind the antibacterial action of ZnO NPs was the suppression of enzymes in the electron transport chain and the generation of ROS by the Zn^2+^ ions released. Additionally, ZnO NPs also destroyed the cell membrane, cell wall, damaged DNA, and induced apoptosis in bacterial cells [51]. In case of a burn wound, ZnO NPs are known to possess antioxidant activity and regulate redox imbalance along with promoting collagen synthesis to repair damaged tissues [52]. Titanium dioxide nanoparticles (TiO_2_ NPs) have also been used in BC to treat burn wounds in mice model. The nanocomposite developed had more than 80% inhibition against both Gram-positive and -negative bacteria. It was observed that this nanocomposite accelerated the wound-healing process through enhanced re-epithelialization and wound contraction [53]. Kamalipooya et al. had used green-synthesized cerium oxide nanoparticles (CeO_2_ NPs) in nanofibers of cellulose acetate to treat diabetic wounds. It was observed that the nanofibers impregnated with CeO_2_ NPs had significant performance against Staphylococcus aureus, and the nanocomposite was able to increase the wound repair rate by more than 95% even after 15 days [54].

Nanoparticles of natural polymers such as chitosan have also been used in the cellulose membranes to design wound dressing materials. In a study by Xia et al., chitosan nanofibers were coated onto cellulose membranes to develop a wound-healing dressing for cutaneous injuries. These nanofibers were incorporated to add functionalities such as the promotion of tissue granulation and cell proliferation during the wound-healing process [55].

In the future, researchers should emphasize developing novel, biocompatible, and green nanoparticles with high antibacterial ability to control the issues of nanotoxicity at the tissue and cellular level.

## 9. Membranes Based on Cellulose Derivatives

Natural cellulose is an abundant material and ideal for membrane fabrication because of its unique porous network structure, mechanical strength, biocompatibility, and environmental friendliness [56]. Cellulose-based multilayer membranes have been used for nanofiltration, gas separation, and reverse osmosis for a long time. The primary challenge of fabricating cellulose-based membranes is to find appropriate solvent material. The interchain hydrogen bonding and strong crystal structure of cellulose make it difficult to dissolve it in all the common organic liquids, including water [57]. The viscose method, or also known as the xanthogenate method, is a conventional technique to dissolve polymer cellulose with sodium hydroxide in carbon disulfide, producing cellulose xanthogenate and a toxic byproduct of hydrogen sulfide [58]. This popular method was coined at the end of the 19th century and has been used to produce viscose rayon fibers for fabricating textiles [59]. Another standard process for producing regenerated cellulose films, fibers, and membranes is called the cuprammonium process, patented by Hoelkeskamp in 1963. This method produces cellulose from the cuprammonium cellulose spinning solution by decomposing the raw cellulose in liquid solution and later extruding it into the required forms [60]. In 1969, a group of US scientists discovered that natural and synthetic polymers with intermolecular solid hydrogen bonding readily dissolve in certain saturated cyclic amine oxides without degrading the compound. The preferred composition is a 70% (by weight) solution of N-methylmorpholine-N-oxide, N-methylpiperidine-N-oxide, N-methylpyrrolidine-N-oxide, and N-methylazacycloheptane-N-oxide [61]. Recently, the N-methylmorpholine-N-oxide (NMMO) route became a more thermally stable and environmentally friendly way of producing cellulose-based films, fibers, and membranes without toxic byproducts [62]. Blow-extruded membranes produced using this technology have advanced properties and can be utilized to develop a wide variety of separation membranes [63].

Multilayer cellulose-based membranes are widely used as an energy-efficient material in water separation and desalination. They can be fabricated using thin cellulose coatings on top of a polymer scaffold. This type of regenerated cellulose membrane is partially crystalline and more hydrophobic than other common membranes. It has a contact angle of less than 22° and high water permeance, thus making it suitable for use in a wide range of solvent mediums [57]. It is observed that the cellulose concentration on the membrane drastically increases the evaporation of liquids on the membrane surface. Glycerin can be used during the cellulose-based membrane regeneration process through a neutralization reaction to control the uniformity of the cellulose concentration on the membrane [63].

Another well-known medium for a homogenous dissolution of cellulose derivatives is ionic liquids (ILs). Ionic liquids have lower melting points and strong hydrogen bond acceptors, found to be able to dissolve cellulose very efficiently [64]. A limitation of this process is the possible degradation of cellulose compounds during the dissolution process. Controlled thermal heating or microwave radiation can be applied to minimize this effect. 1-butyl-3-methylimidazoliumchloride [C4 MIM]Cl and 1-allyl-3-methylimidazoliumchloride [AllylMIM]Cl are the two most popular solvents in use now [65]. However, recent research discovered that 1-butyl3-methylimidazolium chloride provides maximum solubility of cellulose (up to 25 wt%) when controlled thermal heating or radiation is applied [66]. Table 1 presents the dissolution of cellulose in various ILs with strong anions.

The recent development of cellulose synthesis introduces trimethylsilyl cellulose (TMSC) as an alternative to the viscose process. Trimethylsilyl (TMS) groups provide better solubility by reducing the hydrophilic effect and breaking down the strong crystal structure of the cellulose. TMSC quickly decomposes in many organic liquids such as n-hexane and chloroform at relatively low temperatures [67]. In a recent study, Ali developed a high-performance cellulose membrane by applying an acidic hydrolysis of TMSC crosslinked by glyoxal solutions [68]. This crosslinked cellulose was found to have a lower molecular weight cut-off (MWCO), high durability, cost-effectiveness, and suitability for large-scale production. Puspasari et al. [69] later deduced that the crosslinking process of cellulose membrane with glyoxal is reversible and lacks stability. They successfully fabricated a more stable nanofiltration cellulose membrane from TMSC by crosslinking glutaraldehyde instead of glyoxal. The resulting membrane can operate at a much lower pressure than conventional ultrafiltration membranes for desalination and demineralization applications [69].

Biosynthesized membranes based on cellulose derivatives have been widely used in tissue engineering and wound dressing applications in recent years because of their advanced chemical and mechanical properties. Lin et al. [70] produced a large-scale cellulose-chitosan membrane synthesized by *Acetobacter xylinum* that shows low hydration, small pore size, high tensile strength and Young’s modulus, and antibacterial efficacy. They tested the wound dressing properties of the membrane in rat models, which showed a much faster epithelialization and regeneration process in the wound. Xia et al. [55] introduced a one-step, easy method to fabricate chitosan nanofiber-coated porous cellulose membrane by using electrostatic spinning technology. This membrane not only exhibits good antibacterial activity but also excellent light transmittance, wettability, and permeability. Lacin et al. [71] developed antibacterial drug-loaded 2,3-dialdehyde cellulose (DABC)-based hydrogel membranes with excellent biocompatibility and antibacterial efficacy for wound-healing applications. Medical grade chloramphenicol (CAP) was used in this research with a drug loading capacity of 0.1 mg/cm^2^. CAP showed very high drug release efficiency in DABC, as almost 99% of the CAP from the membrane was released within 24 h.

Bacterial cellulose (BC) or microbial cellulose (MC), an advanced biodegradable nanomaterial produced from cellulose derivatives and bacterial cells, is advantageous for therapeutic applications because of its high porosity, purity, and suitability for large-scale production. On top of that, bacterial cellulose can easily be modified, functionalized, and loaded with drugs to enhance its wound dressing responses. Although the mechanical properties of the BCs are dependent on the manufacturing conditions and parameters, their tensile strength and Young’s modulus can be as high as 400 MPa and 114 GPa, respectively [72,73]. The Barrett–Joyner–Halenda model analysis shows that BCs have an average specific surface area of around 60 m^2^/g, average pore volume of around 0.2 cm^3^/g, and average pore size of around 13 nm [20]. BC membranes functionalized by *Gluconacetobacter xylinum* proved to have no cytotoxicity and an excellent wound epithelialization and regeneration capacity [74]. The BC nanomaterial surface can be further doped with metal nanoparticles, such as silver, to increase its healing strength and effectiveness. Silver nanoparticles have intrinsic antimicrobial resistance against many types of multidrug-resistant bacteria [75].

## 10. Applications in Wound Healing

Cellulose-based nanofibers have unique properties in terms of their structure, mechanical strength, and biological interaction. These properties make such nanofibers a promising wound-healing material. The porous scaffolds in these nanofibers resemble the extracellular matrix, and provide support to cells to adhere, migrate, and proliferate. These nanomaterials maintain a moist environment near the wounded site, thereby promoting the healing process [76]. Some of the recent research works highlighting the use of cellulose-based nanofibers in wound healing have been discussed in this section.

Hydrogels containing cellulose nanofibers have recently been used by researchers for wound healing. Wang and co-workers have developed a sprayable hydrogel containing cellulose nanofibers for healing skin wounds. During the research, skin wound models in rat and porcine were selected, and it was observed that the hydrogel helped in regenerating epidermal and dermal tissues along with hair follicles and sebaceous glands [77]. Zhang et al. have also used cellulose nanofibers as reinforcement for developing a hydrogel having antimicrobial, antioxidant, self-healing, and adhesion properties, which are essential for wound healing. In addition to showing excellent antimicrobial activity against Gram-positive and -negative bacteria, the hydrogel also promoted proliferation, angiogenesis, and collagen deposition, thereby signaling towards wound closure. The hydrogel was found to be effective for healing chronic wounds [78].

Researchers have also incorporated various nanoparticles in cellulose nanofibers for added functionalities, further enhancing the wound-healing process. In a study, chitosan nanoparticles encapsulated with cerium oxide nanoparticles were electrospun in cellulose acetate nanofibers to develop a novel dressing material for diabetic wounds. It was observed that this nanocomposite was enhancing the repair of diabetic wounds by 95.47% in a span of 15 days [79]. Similarly, palladium and platinum nanoparticles were also incorporated in the cellulose acetate nanofibers for promoting wound healing. This incorporation helped in producing nanofibers with a very small diameter. This nanocomposite assisted in healing wounds by promoting angiogenesis, granulation, and reducing inflammation [80].

Findings of these studies provide crucial evidence about the wound-healing properties of cellulose nanofibers, and open gates for further research in this field by amalgamating these nanofibers with other biocompatible materials and techniques, thereby further enhancing the wound-healing process.

## 11. Applications in Drug Delivery

The non-bioabsorbable nature of cellulose in the human body has limited the use of cellulose-based membranes as drug-delivery agents via the dermal or oral route. However, modification of these membranes with different biopolymers can assist in controlling and sustaining the release of the loaded drug [1]. In case of wound healing, two major types of drugs are loaded on different cellulose-based wound dressings: antibiotics/antimicrobials and analgesics. These drugs assist the wound dressing in reducing and controlling the infection and pain, respectively, at the wounded site.

In research published by Agarwal et al., membranes of carboxymethyl cellulose (CMC) blended with poly-vinyl alcohol and poly-ethylene oxide were used to deliver the antibiotic, ciprofloxacin hydrochloride. Upon varying the concentration of the polymers blended with the cellulose, the fragility of the membrane was affected. The drug release study showed that it started immediately and carried on till 10 h. Therefore, the researchers suggested that the wound dressing with this composition should be changed every day in order to avoid any chances of infection at the wounded site [2]. Sodium CMC was used by Ng and Jumaat to develop antimicrobial drug-loaded dressings for mucosal wounds. The model drugs chosen for the study were silver nitrate, sulfacetamide sodium, and neomycin trisulfate. Among these drug-loaded CMC membranes, the best wound-healing characteristics were present in the membrane containing neomycin trisulfate. This composite also showed antimicrobial effects against Gram-positive and Gram-negative bacteria [3]. Modified BC membrane with 2,3-dialdehyde has also been used to deliver chloramphenicol. It was observed that this modification helped in controlling the initial burst release and maintaining the drug release for 24 h. While performing the antibacterial studies, it was observed that the composite had antibacterial effects even after 3 days, indicating the sustained antibacterial nature of this wound dressing [4].

Tong et al. utilized cellulose nanocrystals to develop a film loaded with curcumin as an antimicrobial agent for wounds related to diabetes. It was reported that 98.9% of the loaded curcumin was released within 36 h with no burst effect. The wound dressing also showed a broad-spectrum antimicrobial effect and inhibited 99% of microbial growth. The film emerged as a potential agent for diabetic wounds with excellent ability of sebaceous gland and hair follicle regeneration [5]. Cellulose membrane modified with silica particles of mesoporous dimension was used to develop wound dressings containing chloramphenicol as an antimicrobial agent. It was observed that this modification in cellulose membranes helped in releasing the drug in a two-stage process and sustaining it for 270 h. Additionally, the antibacterial activity of this composite was also maintained till 144 h, making this wound dressing a remarkable choice for wounds where frequent dressing changes are not possible [6]. BC, without any further modification, has also been used by Qiu et al. to develop wound-healing dressing material containing vaccarin. This drug is known to promote the growth of epithelial cells. The wound-healing ability of this composite was studied on rat model, and it was found that the epithelialization and regeneration of cells at the wounded site were increased several-fold in the presence of vaccarin released from BC [7].

Analgesics for relieving pain at the wounded site have also been delivered via cellulose membranes. Mayer et al. mimicked the viscose-based wound dressing material by developing a cellulose-based thin film loaded with diclofenac. The drug-loaded film was further spin-coated with additional layers of cellulose to control and decelerate the release of the drug after the initial burst effect. The authors suggested the addition of one of two layers of cellulose above the drug-loaded film to prolong the pain-relieving effect of the developed wound dressing [8]. In another study, Sarkar et al. modified cellulose nanofibrils (prepared from jute fibers) with chitosan to develop a drug-delivery system to sustain the release of ketorolac tromethamine. It was reported that immobilization of the drug with the cellulose nanofibrils helped in controlling its release for 10 h. The researchers suggested the best use of this delivery system via the transdermal route [9].

Though many studies have been published related to the use of cellulose membranes in drug delivery, still a huge scope in the form of surface modifications is available to carry out further developments in this field. Modifications with different nanomaterials, preferably non-toxic and biocompatible, as well as biopolymers, can further enhance the efficiency of these cellulose-based drug-delivery systems. Table 2 summarizes various studies on cellulose-based membranes for drug delivery, highlighting their modifications, loaded drugs, release durations, and applications in wound healing and pain management.

## 12. Applications in Tissue Engineering

Tissue engineering is a broad and multidisciplinary field that combines engineering and life science to repair, restore, and regenerate biological tissues and organs. The primary purpose of this technology is to replace or regrow damaged tissues caused by any disease, abnormalities, or traumatic events [10]. Many biomaterial and hydrogel scaffolds work as seed materials to support cell growth during tissue regeneration. However, natural hydrogels often do not provide enough mechanical strength and support. Surface-modified regenerated cellulose nanofibers enhance mechanical and chemical properties and decompose easily with biological scaffolds for tissue engineering [11]. Hydrogels prepared with native cellulose or cellulose derivatives such as hydroxypropyl methylcellulose (HPMC), methylcellulose (MC), microcrystalline cellulose (MCC), sodium carboxymethylcellulose (NaCMC or CMC), ethylcellulose (EC), etc., are the most common forms of composite material in use. Cellulose derivatives can readily be crosslinked with chitosan, polyvinyl alcohol (PVA), starch, alginates, collagen, etc., for preparing a mechanically robust hydrogel [14]. Nasri-Nasrabadi et al. [14] developed salt-leached starch/cellulose nanofiber scaffolds for cartilage tissue engineering applications. The resulting scaffolds showed good mechanical strength, biodegradability, and uniform pore size, suggesting that Young’s Modulus and tensile strength of the scaffold directly improved with the increase in cellulose nanofiber content, as shown in Table 3. Combined chitosan, PVA, and starch/cellulose hydrogel showed a much higher swelling property (swelling ratio > 234%), flexibility (strain > 300%), and osteoblast tissue cell growth [15].

The surface of the cellulose derivative can be further functionalized with metal nanoparticles to increase the hydrogel’s thermal stability, catalytic activity, and antimicrobial resistance. The chemical crosslinking of copper (Cu), palladium (Pd), silver (Ag), and gold (Au) nanoparticles creates a deposition of metal ions on the surface of the cellulose crystal chain. These metal ions attack microbial cells, creating a microbial-resistant layer adjacent to the damaged tissue cells. The thermogravimetric analysis (TGA) shows that the metal-coated cellulose matrix has excellent thermal stability [16,17]. These advanced properties make crosslinked nanocellulose an ideal material to repair tissues such as heart valve tissue [18], bone tissue [19,20,21], skin tissue [22], cartilage tissue [14,15,16], intervertebral tissue [17], and liver tissue cells [81].

Cellulose derivatives can be prepared from natural resources such as plants, fungi, algae, and bacteria. BC or bacteria nanocellulose (BNC), produced by fermentation of certain bacteria, is another promising material for tissue regeneration applications [19]. British scientist Adrian I. Brown first synthesized bacterial cellulose by fermenting *B. aceti* with vinegar plant-based cellulose [21]. BC covers most cellulose derivatives for tissue engineering applications [22]. Natural *Gluconacetobacter xylinus* is considered the most potent strain for producing bacterial cellulose, as the cellulose synthesized from it exhibits excellent mechanical properties and biodegradability. Unmodified bacterial cellulose shows much higher cartilage cell growth and proliferation of chondrocytes in the human body compared to common tissue culture substrates [21]. S. Torgbo et al. [22] investigated the efficacy of bacterial cellulose for bone tissue regeneration and found that it significantly enhances osseointegration and bone ingrowth. The ease of modification, biocompatibility, microporosity, and the ability to change the chemical structure have made it an ideal scaffold material for osteogenesis and bone tissue engineering. An overview of cellulose-based materials in tissue engineering is shown in Figure 5.

## 13. Future Work

Cellulose-based nanofiber wound dressings have shown great promise in enhancing wound healing, but several areas require further investigation to optimize their performance and clinical applications. Future research should focus on advanced functionalization techniques to improve antimicrobial activity, controlled drug release, and wound-healing efficiency. Developing stimuli-responsive dressings that react to pH, temperature, or enzymatic activity could allow for precise and sustained therapeutic delivery. Additionally, enhancing mechanical properties through composite formulations, hybrid materials, or crosslinking strategies could improve elasticity, durability, and adherence while maintaining biocompatibility.

Another crucial aspect is the clinical translation of these materials. While in vitro and animal studies have demonstrated their effectiveness, extensive clinical trials are needed to validate their safety, efficacy, and long-term impact on wound healing. Regulatory approval processes should also be streamlined to facilitate their commercialization, and scalable, cost-effective manufacturing methods must be developed to ensure accessibility in healthcare settings. The integration of bioengineered cellulose materials with hydrogels, tissue scaffolds, and regenerative medicine presents an exciting avenue for future innovation. Genetically modified cellulose-producing microbes could be explored to fabricate tailored nanofiber structures with enhanced bioactivity. Additionally, evaluating the long-term biodegradability and immune response of cellulose-based dressings in chronic wound environments is essential to ensure their safety and effectiveness. By addressing these challenges, cellulose-based wound dressings can evolve into next-generation therapeutic solutions, offering improved healing, reduced infection rates, and cost-effective treatment options for a wide range of wound care applications.

## Figures and Tables

**Figure 1 biomimetics-10-00344-f001:**
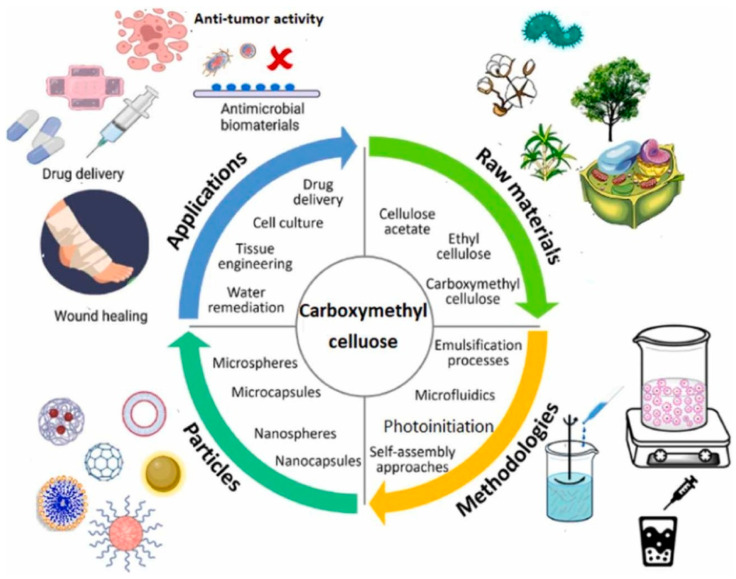
Carboxymethyl cellulose: an overview.

**Figure 2 biomimetics-10-00344-f002:**
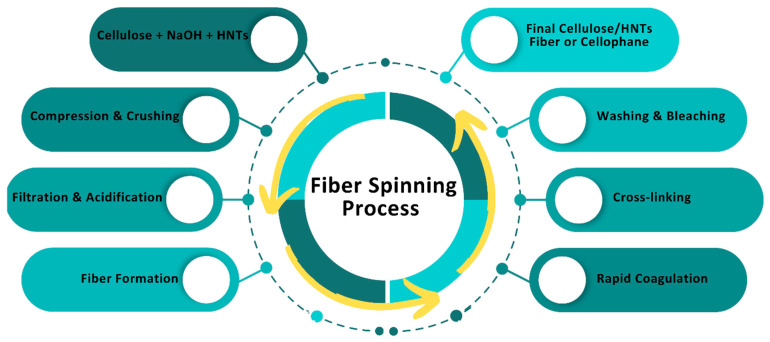
Schematic diagram of the fiber spinning process for the formation of cellulose/HNTs solution.

**Figure 3 biomimetics-10-00344-f003:**
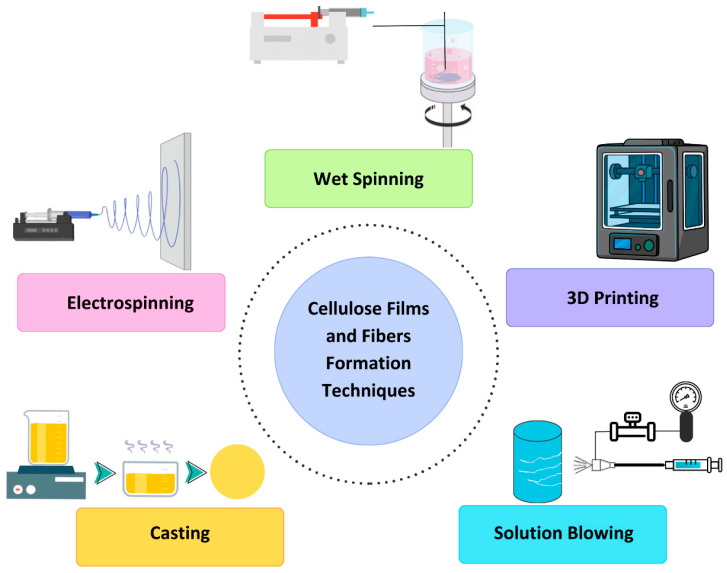
Process flow of cellulose-based film and fiber-formation techniques.

**Figure 4 biomimetics-10-00344-f004:**
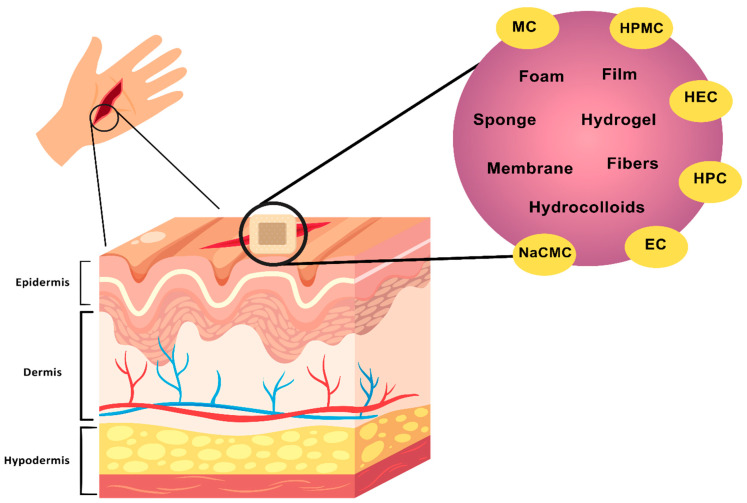
Illustration of the application of cellulose-based wound dressings and their interaction with skin layers. It highlights cellulose derivatives (MC, HPMC, HEC, HPC, NaCMC, EC) and their forms (films, foams, hydrogels, membranes, fibers, sponges, hydrocolloids) used in wound healing. These materials aid in moisture retention, protection, and drug delivery, promoting tissue regeneration, and reducing infection risks.

**Figure 5 biomimetics-10-00344-f005:**
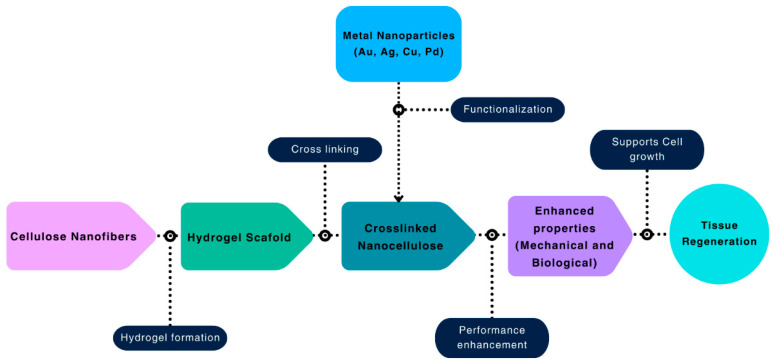
Cellulose-based biomaterials for tissue engineering: from hydrogel formation to tissue regeneration.

**Table 1 biomimetics-10-00344-t001:** Dissolution of cellulose in some common ILs with strong anions.

ILs/Anion	DiMIM	C_2_MIM	C_3_MIM	C_4_MIM	Ally1MIM
F^−^		2 wt% [66]			
Cl^−^		10–14 wt% [66]	No solution [66]	20 wt% [64]	15 wt% [65]
Br^−^		1–2 wt% [66]	1–2 wt% [66]	2–3 wt% [66]	No solution [66]
I^−^	No solution [66]			1–2 wt% [66]	
SCN^−^				5–7 wt% [64]	
TsO^−^		1 wt% [66]			
AcO^−^		8 wt% [66]		12 wt% [66]	
R_2_PO_4_^−^	10 wt% [66]	12–14 wt% [66]		No solution [66]	

DiMIM = 1,2-dimethylimidazolium, C_2_MIM = 1-ethyl-3-methylimidazolium, C_3_MIM = 1-propyl-3-methylimidazolium, C_4_MIM = 1-butyl-3-methylimidazolium, Ally1MIM = 1-allyl-3-methylimidazolium.

**Table 2 biomimetics-10-00344-t002:** Applications of cellulose-based Membranes in drug delivery.

Study/Researcher	Cellulose Modification	Drug Delivered	Release Duration	Application
[2]	CMC blended with PVA and PEO	Ciprofloxacin hydrochloride (antibiotic)	10 h	Daily wound dressing changes are recommended
[3]	Sodium CMC	Silver nitrate, sulphacetamide sodium, neomycin trisulphate	Best effect with neomycin trisulfate	Antimicrobial wound dressing for mucosal wounds
[4]	2,3-dialdehyde modification	Chloramphenicol (antibacterial)	Sustained release for 24 h	Sustained antibacterial effect for 3 days
[5]	Cellulose nanocrystals	Curcumin (antimicrobial)	98.9% release within 36 h	Potential agent for diabetic wound healing
[6]	Silica particles (mesoporous)	Chloramphenicol (antibacterial)	Two-stage release, sustained for 270 h	Long-term antimicrobial dressing (144 h)
[7]	BC without modification	Vaccarin (epithelial growth promoter)	Promoted epithelialization and regeneration	Enhanced wound healing via epithelial growth
[8]	Cellulose-based thin film	Diclofenac (analgesic)	Controlled release with additional cellulose layers	Prolonged pain relief via cellulose layering
[9]	Cellulose nanofibrils with chitosan	Ketorolac tromethamine (analgesic)	Controlled release till 10 h	Transdermal drug-delivery system

**Table 3 biomimetics-10-00344-t003:** Mechanical properties of starch/cellulose nanofiber scaffolds for tissue engineering [15].

Scaffold Sample	Cellulose Nanofiber Content (%)	Salt Content (%)	Young’s Modulus (MPa)	Tensile Strength (MPa)
S/70	0	70	24 ± 5.5	2.6 ± 0.41
SC5/70	5	70	66 ± 3.9	3.61 ± 0.32
SC10/70	10	70	82 ± 7.1	3.82 ± 0.55
SC15/70	15	70	93 ± 3.6	4.03 ± 0.48
S/90	0	90	20 ± 4.2	2.48 ± 0.33
SC5/90	5	90	63 ± 2.9	3.58 ± 0.74
SC10/90	10	90	81 ± 6.5	3.66 ± 0.42
SC15/90	15	90	85 ± 7.5	3.97 ± 0.26

## Data Availability

Data will be made available upon request.

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
