# Peer review of "Cellulose-Based Nanofibers in Wound Dressing"

_biomimetics, 2025, doi:10.3390/biomimetics10060344_

Round 1
Reviewer 1 Report
Comments and Suggestions for Authors
In this review, David K Mills and coworkers present recent focused on the potential of cellulose in wound dressings. (1) structures of cellulose, synthesis methods, formation of films and fibers, cellulose membranes in wound healing and nanoparticle additions for added functionalities based on cellulose were discussed. (2) the potential applications of cellulose in wound healing, drug delivery, tissue engineering fields have been described.
In general, this is a useful work for researchers in this field. I recommend it for publication in Biomimetics after the following concerns are addressed:
- The abstract section is too cumbersome. Please clarify the main idea of the manuscript and reduce the word count.
- Subtitles 8 and 10 are exactly the same, please check and modify them.
- Suggest summarizing the application images in the reference section to provide readers with a more intuitive application effect.
- The format of references needs to be verified: there are many formats for reference formats, please check the entire text and unify the format.
No English issues were detected.
Author Response
Reviewer #1
Comment #1: The abstract section is too cumbersome. Please clarify the main idea of the manuscript and reduce the word count.
Response: As per the suggestion, the abstract is now revised and brought down to 195 words [Line no. 11-28].
Comment #2: Subtitles 8 and 10 are exactly the same, please check and modify them.
Response: The authors are thankful to the reviewer for noticing this mistake. It has now been rectified, and one of the subtitles has been changed [Line no. 272].
Comment #3: Suggest summarizing the application images in the reference section to provide readers with a more intuitive application effect.
Response: The authors are not clear on what action the reviewer is asking us to complete.
Comment #4: The format of references needs to be verified: there are many formats for reference formats, please check the entire text and unify the format.
Response: All references have now been thoroughly reviewed, and any errors have been corrected.
Reviewer 2 Report
Comments and Suggestions for Authors
Dear Authors,
After reading your manuscript, I consider that this one needs major revisions before being published. Below, you will find a list containing few observations:
- Information is not attentively structured. For example: section 8. Membranes Based on Cellulose Derivatives, and section 10: Membranes Based on Cellulose Derivatives. The same title, different information?
-
Sections 4 and 5 together, maybe. What the difference between the subjects treated by these two sections?
-
Sections 6.1, and 6.2 - too succinct presented, in the conditions in which these techniques are considered to be “prominent strategies owing to their efficiency and versatility”, between all the techniques that can be used for surface functionalization of cellulose membranes
-
Section 11 – only general information on using cellulose for wound healing (11 lines), even if the title of this paper is about this subject. No examples on using cellulose in this application field.
-
Even if the title of the article is “Cellulose-Based Nanofibers in Wound Dressing”, not very much information on these nanosystems
-
Line 77-78: “For the development of ‘smart’ wound dressings, natural polymers”…..phrase is not finished. I think a part of the text is missing, because there is no information on Figure 1, in the text.
-
No references for Figures 1, 2, 3, 4, 5, next to the Figure caption
- Figures 2, 4, 5 are not introduced and explained in the text. They just appear as Figures, without referring to them in the text
-
Only the acronym for MDR bacteria (line 480) – in the text, the whole name for these acronyms have to be mentioned – multidrug resistant bacteria
-
Ref for Table 1 is missing
-
Lines 562, 566 – the references are correct?
-
Line 576: ….., as shown in Table 1”. – Table 3, instead of Table 1
- The English is pretty fine, but it is better to check its correctness once again; for ex, in line 343: ….the composite was having significant antibacterial activity…
-
There is no uniformity in writing the references. For example, they are written either as “…Cellulose 30, 2037–2052 (2023)” (ref 29), either as “…2013(1), 417672” (ref 24)
-
Reference 11. https://www.mdpi.com/1420- 661 3049/25/21/5097 - File not found, and not written as the other reference
Author Response
Subject: Response to Reviewers’ Comments (Manuscript ID: biomimetics-3592787)
Reviewer #2:
Comment #1: Information is not attentively structured. For example: section 8. Membranes Based on Cellulose Derivatives, and section 10: Membranes Based on Cellulose Derivatives. The same title, different information?
Response: The authors have now rectified this mistake, and one of the subtitles has been changed [Line no. 272].
Comment #2: Sections 4 and 5 together, maybe. What the difference between the subjects treated by these two sections?
Response: As per the valuable suggestion, both the sections have now been merged [Line no. 120-189].
Comment #3: Sections 6.1, and 6.2 - too succinct presented, in the conditions in which these techniques are considered to be “prominent strategies owing to their efficiency and versatility”, between all the techniques that can be used for surface functionalization of cellulose membranes
Response: In our submitted manuscript, there are no Sections 6.1, and 6.2.
Comment #4: Section 11 – only general information on using cellulose for wound healing (11 lines), even if the title of this paper is about this subject. No examples on using cellulose in this application field.
Response: We are thankful to the reviewer for noticing this mistake. We have now rewritten this section and included recent research works relevant to the subtitle [Line no. 472-504].
Comment #5: Even if the title of the article is “Cellulose-Based Nanofibers in Wound Dressing”, not very much information on these nanosystems.
Response: The entire manuscript discusses cellulose nanofibers, their functionalization, application in various fabrication methods, and their incorporation into membranes.
Comment #6: Line 77-78: “For the development of ‘smart’ wound dressings, natural polymers”…..phrase is not finished. I think a part of the text is missing, because there is no information on Figure 1, in the text.
Response: The phrase mentioned here has now been removed, and Figure 1 has now been cited in the relevant text.
Comment #7: No references for Figures 1, 2, 3, 4, 5, next to the Figure caption
Response: We would like to respectfully inform the reviewer that all figures included in the manuscript have been originally created by the authors and have not been adapted from any previous publications.
Comment #8: Figures 2, 4, 5 are not introduced and explained in the text. They just appear as Figures, without referring to them in the text
Response: We thank the reviewer for noticing this mistake. All the figures have now been cited next to the relevant text in the manuscript.
Comment #9: Only the acronym for MDR bacteria (line 480) – in the text, the whole name for these acronyms has to be mentioned – multidrug resistant bacteria
Response: As per the suggestion, the whole name of MDR is now mentioned in the text [Line no. 470-471].
Comment #10: Ref for Table 1 is missing
Response: Relevant references have now been added in table 1.
Comment #11: Lines 562, 566 – the references are correct?
Response: We sincerely thank the reviewer for such careful observation. The references were correct, however, they were written in superscript. They have now been written as per the journal’s format.
Comment #12: Line 576: ….., as shown in Table 1”. – Table 3, instead of Table 1
Response: This mistake has now been rectified [Line no. 585].
Comment #13: The English is pretty fine, but it is better to check its correctness once again; for ex, in line 343: ….the composite was having significant antibacterial activity…
Response: The entire manuscript has now been thoroughly checked for any grammatical mistakes and typos.
Comment #14: There is no uniformity in writing the references. For example, they are written either as “…Cellulose 30, 2037–2052 (2023)” (ref 29), either as “…2013(1), 417672” (ref 24)
Response: All references have now been thoroughly reviewed, and any errors have been corrected.
Comment #15: Reference 11. https://www.mdpi.com/1420- 661 3049/25/21/5097 - File not found, and not written as the other reference
Response: The mistake mentioned here has now been rectified.
Reviewer 3 Report
Comments and Suggestions for Authors
This manuscript, “Cellulose-Based Nanofibers in Wound Dressing” presents a comprehensive review of the potential of cellulose and its derivatives in wound dressing applications. It thoroughly discusses the chemical structure, processing strategies, functionalization approaches, and nanoparticle-based enhancements of cellulose while aligning these aspects with the practical requirements of wound care. The review is well-organized and supported by extensive references, offering valuable contributions to the field. Thus, it is recommended for publication in the Journal of Biomimetics following minor revisions:
- It is suggested that more graphs (for example, a schematic illustration of the experimental data or the preparation process) could be added to provide the reader with a more graphic comprehension of the procedure.
- There are inconsistencies in the formatting of some of the literature, such as references 4, 6, 18, etc., and it is recommended that they be reviewed and adjusted to a standardized citation format.
- Some parts are too redundant with excessive information, and the language could be appropriately simplified to improve readability.
Author Response
Reviewer #3
Comment #1: It is suggested that more graphs (for example, a schematic illustration of the experimental data or the preparation process) could be added to provide the reader with a more graphic comprehension of the procedure.
Response: Figures 2 and 3 graphically depict data included from previous publications and a graphic that summarized current fabrication methods. We have also created a graphical abstract of current preparation process).
Comment #2: There are inconsistencies in the formatting of some of the literature, such as references 4, 6, 18, etc., and it is recommended that they be reviewed and adjusted to a standardized citation format.
Response: All the errors in the references section have now been rectified.
Comment #3: Some parts are too redundant with excessive information, and the language could be appropriately simplified to improve readability.
Response: As per the valuable suggestion, we have improved the readability of the manuscript.